# Online Learning of Optimal Bidding Strategy in Repeated Multi-Commodity Auctions

**Sevi Baltaoglu**
Cornell University
Ithaca, NY 14850
msb372@cornell.edu

**Lang Tong**
Cornell University
Ithaca, NY 14850
lt35@cornell.edu

**Qing Zhao**
Cornell University
Ithaca, NY 14850
qz16@cornell.edu

## Abstract

We study the online learning problem of a bidder who participates in repeated auctions. With the goal of maximizing his T-period payoff, the bidder determines the optimal allocation of his budget among his bids for $K$ goods at each period. As a bidding strategy, we propose a polynomial-time algorithm, inspired by the dynamic programming approach to the knapsack problem. The proposed algorithm, referred to as dynamic programming on discrete set (DPDS), achieves a regret order of $O(\sqrt{T \log T})$. By showing that the regret is lower bounded by $\Omega(\sqrt{T})$ for any strategy, we conclude that DPDS is order optimal up to a $\sqrt{\log T}$ term. We evaluate the performance of DPDS empirically in the context of virtual trading in wholesale electricity markets by using historical data from the New York market. Empirical results show that DPDS consistently outperforms benchmark heuristic methods that are derived from machine learning and online learning approaches.

## 1 Introduction

We consider the problem of optimal bidding in a multi-commodity uniform-price auction (UPA) [1], which promotes the law of one price for identical goods. UPA is widely used in practice. Examples include spectrum auction, the auction of treasury notes, the auction of emission permits (UK), and virtual trading in the wholesale electricity market, which we discuss in detail in Sec. 1.1.

A mathematical abstraction of multi-commodity UPA is as follows. A bidder has $K$ goods to bid on at an auction. With the objective to maximize his T-period expected profit, at each period, the bidder determines how much to bid for each good subject to a budget constraint.

In the bidding period $t$, if a bid $x_{t,k}$ for good $k$ is greater than or equal to its *auction clearing price* $\lambda_{t,k}$, then the bid is cleared, and the bidder pays $\lambda_{t,k}$. His revenue resulting from the cleared bid will be the good's *spot price* (utility) $\pi_{t,k}$. In particular, the payoff obtained from good $k$ at period $t$ is $(\pi_{t,k} - \lambda_{t,k})\mathbb{1}\{x_{t,k} \geq \lambda_{t,k}\}$ where $\mathbb{1}\{x_{t,k} \geq \lambda_{t,k}\}$ indicates whether the bid is cleared. Let $\lambda_t = [\lambda_{t,1}, ..., \lambda_{t,K}]^\mathsf{T}$ and $\pi_t = [\pi_{t,1}, ..., \pi_{t,K}]^\mathsf{T}$ be the vector of auction clearing and spot market prices at period $t$, respectively. Similarly, let $x_t = [x_{t,1}, ..., x_{t,K}]^\mathsf{T}$ be the vector of bids for period $t$. We assume that $(\pi_t, \lambda_t)$ are drawn from an unknown joint distribution and, in our analysis, independent and identically distributed (i.i.d.) over time.[1]

At the end of each period, the bidder observes the auction clearing and spot prices of all goods. Therefore, before choosing the bid of period $t$, all the information the bidder has is a vector $I_{t-1}$ containing his observation and decision history $\{x_i, \lambda_i, \pi_i\}_{i=1}^{t-1}$. Consequently, a bidding policy $\mu$ of a bidder is defined as a sequence of decision rules, *i.e.*, $\mu = (\mu_0, \mu_1..., \mu_{T-1})$, such that, at time $t-1$,

$\mu_{t-1}$ maps the information history $I_{t-1}$ to the bid $x_t$ of period $t$. The performance of any bidding policy $\mu$ is measured by its regret, which is defined by the difference between the total expected payoff of policy $\mu$ and that of the optimal bidding strategy under known distribution of $(\pi_t, \lambda_t)$.

## 1.1 Motivating applications

The mathematical abstraction introduced above applies to virtual trading in the U.S. wholesale electricity markets that are operated under a two-settlement framework. In the day-ahead (DA) market, the independent system operator (ISO) receives offers to sell and bids to buy from generators and retailers for each hour of the next day. To determine the optimal DA dispatch of the next day and DA electricity prices at each location, ISO solves an economic dispatch problem with the objective of maximizing social surplus while taking transmission and operational constraints into account. Due to system congestion and losses, wholesale electricity prices vary from location to location.[2] In the real-time (RT) market, ISO adjusts the DA dispatch according to the RT operating conditions, and the RT wholesale price compensates deviations in the actual consumption from the DA schedule.

The differences between DA and RT prices occur frequently both as a result of generators and retailers exercising locational market power [2] and as a result of price spikes in the RT due to unplanned outages and unpredictable weather conditions [3]. To promote price convergence between DA and RT markets, in the early 2000s, virtual trading was introduced [4]. Virtual trading is a financial mechanism that allows market participants and external financial entities to arbitrage on the differences between DA and RT prices. Empirical and analytical studies have shown that increased competition in the market due to virtual trading results in price convergence and increased market efficiency [2, 3, 5].

Virtual transactions make up a significant portion of the wholesale electricity markets. For example, the total volume of cleared virtual transactions in five big ISO markets was 13% of the total load in 2013 [4]. In the same year, total payoff resulting from all virtual transactions was around 250 million dollars in the PJM market [2] and 45 million dollars in NYISO market [6].

A bid in virtual trading is a bid to buy (sell) energy in the DA market at a specific location with an obligation to sell (buy) back exactly the same amount in the RT market at the same location if the bid is cleared (accepted). Specifically, a bid to buy in the DA market is cleared if the offered bid price is higher than the DA market price. Similarly, a bid to sell in the DA market is cleared if it is below the DA market price. In this context, different locations and/or different hours of the day are the set of goods to bid on. The DA prices are the auction clearing prices, and the RT prices are the spot prices.

The problem studied here may also find applications in other types of repeated auctions where the auction may be of the double, uniform-price, or second-price types. For example, in the case of online advertising auctions [7], different goods can correspond to different types of advertising space an advertiser may consider to bid on.

## 1.2 Main results and related work

We propose an online learning approach to the algorithmic bidding under budget constraints in repeated multi-commodity auctions. The proposed approach falls in the category of empirical risk minimization (ERM) also referred to as the follow the leader approach. The main challenge here is that optimizing the payoff (risk) amounts to solving a multiple choice knapsack problem (MCKP) that is known to be NP hard [8]. The proposed approach, referred to as dynamic programming on discrete set (DPDS), is inspired by a pseudo-polynomial dynamic programming approach to 0-1 Knapsack problems. DPDS allocates the limited budget of the bidder among $K$ goods in polynomial time both in terms of the number of goods $K$ and in terms of the time horizon $T$. We show that the expected payoff of DPDS converges to that of the optimal strategy under known distribution by a rate no slower than $\sqrt{\log t / t}$ which results in a regret upper bound of $O(\sqrt{T \log T})$. By showing that, for any bidding strategy, the regret is lower bounded by $\Omega(\sqrt{T})$, we prove that DPDS is order optimal up to a $\sqrt{\log T}$ term. We also evaluate the performance of DPDS empirically in the context of virtual trading by using historical data from the New York energy market. Our empirical results show that

DPDS consistently outperforms benchmark heuristic methods that are derived from standard machine learning methods.

The problem formulated here can be viewed in multiple machine learning perspectives. We highlight below several relevant existing approaches. Since the bidder can calculate the reward that could have been obtained by selecting any given bid value regardless of its own decision, our problem falls into the category of *full-feedback* version of multi-armed bandit (MAB) problem, referred to as *experts* problem, where the reward of all arms (actions) are observable at the end of each period regardless of the chosen arm. For the case of finite number of arms, Kleinberg et al. [9] showed that, for stochastic setting, constant regret is achievable by choosing the arm with the highest average reward at each period. A special case of the adversarial setting was studied by Cesa-Bianchi et al. [10] who provided matching upper and lower bounds in the order of $\Theta(\sqrt{T})$. Later, Freund and Schapire [11] and Auer et al. [12] showed that the Hedge algorithm, a variation of weighted majority algorithm [13], achieves the matching bound for the general setting. These results, however, do not apply to experts problems with continuous action spaces.

The stochastic experts problem where the set of arms is an uncountable compact metric space $(\mathcal{X}, d)$ rather than finite was studied by Kleinberg and Slivkins [14] (see [15] for an extended version). Since there are uncountable number of arms, it is assumed that, in each period, a payoff function drawn from an i.i.d. distribution is observed rather than the individual payoff of each arm. Under the assumption of Lipschitz expected payoff function, they showed that the instance-specific regret of any algorithm is lower bounded by $\Omega(\sqrt{T})$. They also showed that their algorithm—NaiveExperts—achieves a regret upper bound of $O(T^{\gamma})$ for any $\gamma > (b+1)/(b+2)$ where $b$ is the isometry invariant of the metric space. However, NaiveExperts is computationally intractable in practice because the computational complexity of its direct implementation grows exponentially with the dimension (number of goods in our case). Furthermore, the lower bound in [14] does not imply a lower bound for our problem with a specific payoff. Krichene et al. [16] studied the adversarial setting and proposed an extension of the Hedge algorithm, which achieves $O(\sqrt{T \log T})$ regret under the assumption of Lipschitz payoff functions. For our problem, it is reasonable to assume that the expected payoff function is Lipschitz; yet it is clear that, at each period, the payoff realization is a step function which is not Lipschitz. Hence, Lipschitz assumption of [16] doesn't hold in our setting.

Stochastic gradient descent methods, which have low computational complexity, have been extensively studied in the literature of continuum-armed bandit [17, 18, 19]. However, either the concavity or the unimodality of the expected payoff function is required for regret guarantees of these methods to hold. This may not be the case in our problem depending on the underlying distribution of prices.

A relevant work that takes an online learning perspective for the problem of a bidder engaging in repeated auctions is Weed et al. [7]. They are motivated by online advertising auctions and studied the partial information setting of the same problem as ours but without a budget constraint. Under the margin condition, *i.e.*, the probability of auction price occurring in close proximity of mean utility is bounded, they showed that their algorithm, inspired by the UCB1 algorithm [20], achieves regret that ranges from $O(\log T)$ to $O(\sqrt{T \log T})$ depending on how tight the margin condition is. They also provided matching lower bounds up to a logarithmic factor. However, their lower bound does not imply a bound for the full information setting we study here. Also, the learning algorithm in [7] does not apply here because the goods are coupled through the budget constraint in our case. Furthermore, we do not have margin condition, and we allow the utility of the good to depend on the auction price.

Some other examples of literature on online learning in repeated auctions studied the problem of an advertiser who wants to maximize the number of clicks with a budget constraint [21, 22], or that of a seller who tries to learn the valuation of its buyer in a posted price auction [23, 24]. The settings considered in those problems are considerably different from that studied here in the implementation of budget constraints [21, 22], and in the strategic behavior of the bidder [23, 24].

## 2   Problem formulation

The total expected payoff at period $t$ given bid $x_t$ can be expressed as

$$r(x_t) = \mathbb{E}\left((\pi_t - \lambda_t)^{\mathsf{T}} \mathbb{1}\{x_t \geq \lambda_t\} | x_t\right),$$

where the expectation is taken using the joint distribution of $(\pi_t, \lambda_t)$, and $\mathbb{1}\{x_t \geq \lambda_t\}$ is the vector of indicator functions with the $k$-th entry corresponding to $\mathbb{1}\{x_{t,k} \geq \lambda_{t,k}\}$. We assume that the payoff

$(\pi_t - \lambda_t)^{\mathsf{T}} \mathbb{1}\{x_t \geq \lambda_t\}$ obtained at each period is a bounded random variable with support in $[l, u]$,[3] and the auction prices are drawn from a distribution with positive support. Hence, a zero bid for any good is equivalent to not bidding because it will not get cleared.

The objective is to determine a bidding policy $\mu$ that maximizes the expected T-period payoff subject to a budget constraint for each individual period:

$$
\begin{aligned}
\underset{\mu}{\text{maximize}} \quad & \mathbb{E}\left(\sum_{t=1}^{T} r(x_t^{\mu})\right) \\
\text{subject to} \quad & \|x_t^{\mu}\|_1 \leq B, \qquad \text{for all } t = 1, ..., T, \\
& x_t^{\mu} \geq 0, \qquad \text{for all } t = 1, ..., T,
\end{aligned}
\tag{1}
$$

where $B$ is the auction budget of the bidder, $x_t^{\mu}$ denotes the bid determined by policy $\mu$, and $x_t^{\mu} \geq 0$ is equivalent to $x_{t,k}^{\mu} \geq 0$ for all $k \in \{1, 2, ..., K\}$.

## 2.1 Optimal solution under known distribution

If the joint distribution $f(.,.)$ of $\pi_t$ and $\lambda_t$ is known, the optimization problem (1) decouples to solving for each time instant separately. Since $(\pi_t, \lambda_t)$ is i.i.d. over $t$, an optimal solution under known model does not depend on $t$ and is given by

$$
x^* = \underset{x_t \in \mathcal{F}}{\arg\max} \, r(x_t)
\tag{2}
$$

where $\mathcal{F} = \{x \in \Re^K : x \geq 0, \|x\|_1 \leq B\}$ is the feasible set of bids. Optimal solution $x^*$ may not be unique or it may not have a closed form. The following example illustrates a case where there isn't a closed form solution and shows that, even in the case of known distribution, the problem is a combinatorial stochastic optimization, and it is not easy to calculate an optimal solution.

**Example.** Let $\lambda_t$ and $\pi_t$ be independent, $\lambda_{t,k}$ be exponentially distributed with mean $\bar{\lambda}_k > 0$, and the mean of $\pi_{t,k}$ be $\bar{\pi}_k > 0$ for all $k \in \{1, .., K\}$. Since not bidding for good $k$ is optimal if $\bar{\pi}_k \leq 0$, we exclude the case $\bar{\pi}_k \leq 0$ without loss of generality. For this example, we can use the concavity of $r(x)$ in the interval $[0, \bar{\pi}]$, where $\bar{\pi} = [\bar{\pi}_1, ..., \bar{\pi}_K]^{\mathsf{T}}$, to obtain the unique optimal solution $x^*$, which is characterized by

$$
x_k^* = \begin{cases}
\bar{\pi}_k & \text{if } \sum_{k=1}^{K} \bar{\pi}_k \leq B, \\
0 & \text{if } \sum_{k=1}^{K} \bar{\pi}_k > B \text{ and } \bar{\pi}_k/\bar{\lambda}_k < \gamma^*, \\
x_k \text{ satisfying } (\bar{\pi}_k - x_k)e^{-x_k/\bar{\lambda}_k}/\bar{\lambda}_k = \gamma^* & \text{if } \sum_{k=1}^{K} \bar{\pi}_k > B \text{ and } \bar{\pi}_k/\bar{\lambda}_k \geq \gamma^*,
\end{cases}
$$

where the Lagrange multiplier $\gamma^* > 0$ is chosen such that $\|x^*\|_1 = B$ is satisfied. This solution takes the form of a "water-filling" strategy. More specifically, if the budget constraint is not binding, then the optimal solution is to bid $\bar{\pi}_k$ for every good $k$. However, in the case of a binding budget constraint, the optimal solution is determined by the bid value at which the marginal expected payoff associated with each good $k$ is equal to $\min(\gamma^*, \bar{\pi}_k/\bar{\lambda}_k)$, and this bid value cannot be expressed in closed form.

We measure the performance of a bidding policy $\mu$ by its regret[4], the difference between the expected T-period payoff of $\mu$ and that of $x^*$, *i.e.*,

$$
\mathcal{R}_T^{\mu}(f) = \sum_{t=1}^{T} \mathbb{E}(r(x^*) - r(x_t^{\mu})),
\tag{3}
$$

where the expectation is taken with respect to the randomness induced by $\mu$. The regret of any policy is monotonically increasing. Hence, we are interested in policies with sub-linear regret growth.

# 3 Online learning approach to optimal bidding

The idea behind our approach is to maximize the sample mean of the expected payoff function, which is an ERM approach [26]. However, we show that a direct implementation of ERM is NP-hard. Hence, we propose a polynomial-time algorithm that is based on dynamic programming on a discretized feasible set. We show that our approach achieves the order optimal regret.

## 3.1 Approximate expected payoff function and its optimization

Regardless of the bidding policy, one can observe the auction and spot prices of past periods. Therefore, the average payoff that could have been obtained by bidding $x$ up to the current period can be calculated for any fixed value of $x \in \mathcal{F}$. Specifically, the average payoff $\hat{r}_{t,k}(x_k)$ for a good $k$ as a function of the bid value $x_k$ can be calculated at period $t+1$ by using observations up to $t$, *i.e.,*

$$\hat{r}_{t,k}(x_k) = (1/t)\sum_{i=1}^{t}(\pi_{i,k} - \lambda_{i,k})\mathbb{1}\{x_k \geq \lambda_{i,k}\}.$$

For example, at the end of first period, $\hat{r}_{t,k}(x_k) = (\pi_{1,k} - \lambda_{1,k})\mathbb{1}\{x_k \geq \lambda_{1,k}\}$ as illustrated in Fig. 1a. For, $t \geq 2$, this can be expressed recursively;

$$\hat{r}_{t,k}(x_k) = \begin{cases} \frac{t-1}{t}\hat{r}_{t-1,k}(x_k) & \text{if } x_k < \lambda_{t,k}, \\ \frac{t-1}{t}\hat{r}_{t-1,k}(x_k) + \frac{1}{t}(\pi_{t,k} - \lambda_{t,k}) & \text{if } x_k \geq \lambda_{t,k}. \end{cases} \quad (4)$$

Since each observation introduces a new breakpoint, and the value of average payoff function is constant between two consecutive breakpoints, we observe that $\hat{r}_{t,k}(x_k)$ is a piece-wise constant function with at most $t$ breakpoints. Let the vector of order statistics of the observed auction clearing prices $\{\lambda_{i,k}\}_{i=1}^{t}$ and zero be $\lambda^{(k)} = \left[0, \lambda_{(1),k}, ..., \lambda_{(t),k}\right]^{\mathsf{T}}$, and let the vector of associated average payoffs be $r^{(k)}$, *i.e.,* $r_i^{(k)} = \hat{r}_{t,k}\left(\lambda_i^{(k)}\right)$. Then, $\hat{r}_{t,k}(x_k)$ can be expressed by the pair $\left(\lambda^{(k)}, r^{(k)}\right)$, *e.g.,* see Fig. 1b.

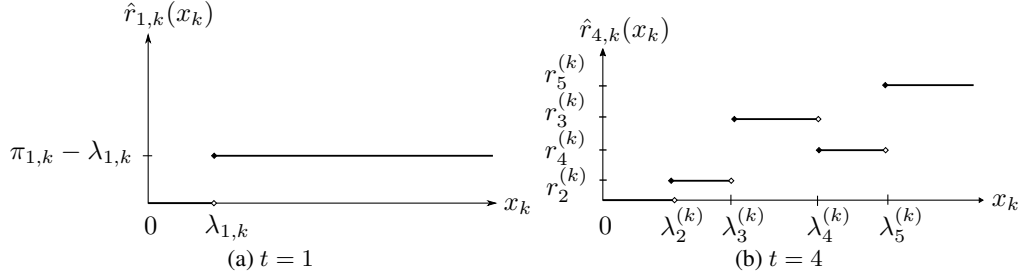

Figure 1: Piece-wise constant average payoff function of good $k$

For a vector $y$, let $y_{m:n} = (y_m, y_{m+1}, ..., y_n)$ denote the sequence of entries from $m$ to $n$. Initialize $\left(\lambda^{(k)}, r^{(k)}\right) = (0, 0)$ at the beginning of first period. Then, at each period $t \geq 1$, the pair $\left(\lambda^{(k)}, r^{(k)}\right)$ can be updated recursively as follows:

$$\left(\lambda^{(k)}, r^{(k)}\right) = \left(\left[\lambda_{1:i_k}^{(k)}, \lambda_{t,k}, \lambda_{i_k+1:t}^{(k)}\right]^{\mathsf{T}}, \left[\frac{t-1}{t}r_{1:i_k}^{(k)}, \frac{t-1}{t}r_{i_k:t}^{(k)} + \frac{1}{t}(\pi_{t,k} - \lambda_{t,k})\right]^{\mathsf{T}}\right), \quad (5)$$

where $i_k = \max_{i:\lambda_i^{(k)} < \lambda_{t,k}} i$ at period $t$.

Consequently, overall average payoff function $\hat{r}_t(x)$ can be expressed as a sum of average payoff functions of individual goods. Instead of the unknown expected payoff $r(x)$, let's consider the maximization of the average payoff function, which corresponds to the ERM approach, *i.e.,*

$$\max_{x \in \mathcal{F}} \hat{r}_t(x) = \max_{x \in \mathcal{F}} \sum_{k=1}^{K} \hat{r}_{t,k}(x_k). \quad (6)$$

Due to the piece-wise constant structure, choosing $x_k = \lambda_i^{(k)}$ for some $i \in \{1, ..., t+1\}$ contributes the same amount to the overall payoff as choosing any $x_k \in \left[\lambda_i^{(k)}, \lambda_{i+1}^{(k)}\right)$ if $i < t+1$ and any

$x_k \geq \lambda_i^{(k)}$ if $i = t+1$. However, choosing $x_k = \lambda_i^{(k)}$ utilizes a smaller portion of the budget. Hence, an optimal solution to (6) can be obtained by solving the following integer linear program:

$$
\begin{aligned}
\underset{\{z_k\}_{k=1}^K}{\text{maximize}} \quad & \sum_{k=1}^{K} \left( r^{(k)} \right)^\mathsf{T} z_k \\
\text{subject to} \quad & \sum_{k=1}^{K} \left( \lambda^{(k)} \right)^\mathsf{T} z_k \leq B, \\
& \mathbf{1}^\mathsf{T} z_k \leq 1, \qquad \forall k = 1, ..., K, \\
& z_{k,i} \in \{0,1\}, \quad \forall i = 1, ..., t+1; \forall k = 1, ..., K.
\end{aligned}
\tag{7}
$$

where the bid value $x_k = \left( \lambda^{(k)} \right)^\mathsf{T} z_k$ for good $k$.

Observe that (7) is a multiple choice knapsack problem (MCKP) [8], a generalization of 0-1 knapsack. Unfortunately, (7) is NP-hard [8]. If we had a polynomial-time algorithm that finds an optimal solution $x \in \mathcal{F}$ to (6), then we could have obtained the solution of (7) in polynomial-time too by setting $z_{k,i} = 1$ where $i = \max_{i:\lambda_i^{(k)} \leq x_k} i$ for each $k$. Therefore, (6) is also NP-hard, and, to the best of our knowledge, there isn't any method in the ERM literature [27], which mostly focuses on classification problems, suitable to implement for the specific problem at hand.

## 3.2 Dynamic programming on discrete set (DPDS) policy

Next, we present an approach that discretizes the feasible set using intervals of equal length and optimizes the average payoff on this new discrete set via a dynamic program. Although this approach doesn't solve (6), the solution can be arbitrarily close to the optimal depending on the choice of the interval length under the assumption of the Lipschitz continuous expected payoff function. To exploit the smoothness of Lipschitz continuity, discretization approach of the continuous feasible set has been used in the continuous MAB literature previously [17, 14]. However, different than MAB literature, in this paper, discretization approach is utilized to reduce the computational complexity of an NP-hard problem as well.

Let $\alpha_t$ be an integer sequence increasing with $t$ and $\mathcal{D}_t = \{0, B/\alpha_t, 2B/\alpha_t, ..., B\}$ as illustrated in Fig. 2. Then, the new discrete set is given as $\mathcal{F}_t = \{x \in \mathcal{F} : x_k \in \mathcal{D}_t, \forall k \in \{1, ..., K\}\}$. Our goal is to optimize $\hat{r}_t(.)$ on the new set $\mathcal{F}_t$ rather than $\mathcal{F}$, *i.e.*,

$$
\max_{x_{t+1} \in \mathcal{F}_t} \hat{r}_t(x_{t+1}).
\tag{8}
$$

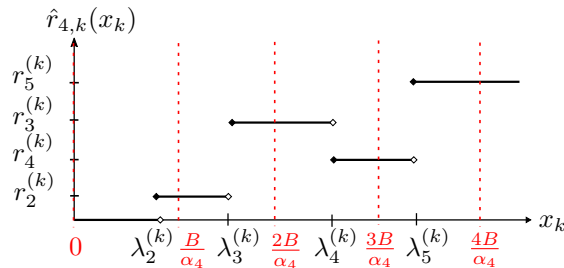

Figure 2: Example of the discretization of the decision space for good $k$ when $t = 4$

Now, we use dynamic programming approach that has been used to solve 0-1 Knapsack problems including MCKP given in (7) [28]. However, direct implementation of this approach results in pseudo-polynomial computational complexity in the case of 0-1 Knapsack problems. The discretization of the feasible set with equal interval length reduces the computational complexity to polynomial time.

We define the maximum payoff one can collect with budget $b$ among goods $\{1, ..., n\}$ when the bid value $x_k$ is restricted to the set $\mathcal{D}_t$ for each good $k$ as

$$
V_n(b) = \max_{\{x_k\}_{k=1}^n : \sum_{k=1}^n x_k \leq b, x_k \in \mathcal{D}_t \forall k} \sum_{k=1}^{n} \hat{r}_{t,k}(x_k).
$$

Then, the following recursion can be used to solve for $V_K(B)$ which gives the optimal solution to (8):

$$V_n(jB/\alpha_t) = \begin{cases} 0 & \text{if } n = 0, j \in \{0, 1, ..., \alpha_t\}, \\ \max_{0 \le i \le j} (\hat{r}_{t,n}(iB/\alpha_t) + V_{n-1}((j-i)B/\alpha_t)) & \text{if } 1 \le n \le K, j \in \{0, 1, ..., \alpha_t\}. \end{cases} \quad (9)$$

This is the Bellman equation where $V_n(b)$ is the maximum total payoff one can collect using remaining budget $b$ and remaining $n$ goods. Its optimality can be shown via a simple induction argument. Recall that $\hat{r}_{t,n}(0) = 0$ for all $(t, n)$ pairs due to the assumption of positive day-ahead prices.

Recursion (9) can be solved starting from $n = 1$ and proceeding to $n = K$, where, for each $n$, $V_n(b)$ is calculated for all $b \in \mathcal{D}_t$. Since the computation of $V_n(b)$ requires at most $\alpha_t + 1$ comparison for any fixed value of $n \in \{1, ..., K\}$ and $b \in \mathcal{D}_t$, it has a computational complexity on the order of $K\alpha_t^2$ once the average payoff values $\hat{r}_{t,n}(x_n)$ for all $x_n \in \mathcal{D}_t$ and $n \in \{1, ..., K\}$ are given. For each $n \in \{1, ..., K\}$, computation of $\hat{r}_{t,n}(x_n)$ for all $x_n \in \mathcal{D}_t$ introduces an additional computational complexity of at most on the order of $t$, which can be observed from the update step of $(\lambda^{(k)}, \pi^{(k)})$, given in (5). Hence, total computational complexity of DPDS is $O(K \max(t, \alpha_t^2))$ at each period $t$.

### 3.3 Convergence and regret of DPDS policy

Under the assumption of Lipschitz continuity, Theorem 1 shows that the value of DPDS converges to the value of the optimal policy under known model with a rate faster than or equal to $\sqrt{\log t / t}$ if the DPDS algorithm parameter $\alpha_t = \lceil t^\gamma \rceil$ with $\gamma \ge 1/2$. Consequently, the regret growth rate of DPDS is upper bounded by $O(\sqrt{T \log T})$. If $\gamma = 1/2$, then the computational complexity of the algorithm is bounded by $O(Kt)$ at each period $t$, and total complexity over the entire horizon is $O(KT^2)$.

**Theorem 1** *Let $x_{t+1}^{DPDS}$ denote the bid of DPDS policy for period $t + 1$. If $r(.)$ is Lipschitz continuous on $\mathcal{F}$ with p-norm and Lipschitz constant $L$, then, for any $\gamma > 0$ and for DPDS parameter choice $\alpha_t \ge 2$,*

$$\mathbb{E}(r(x^*) - r(x_{t+1}^{DPDS})) \le \frac{LK^{1/p}B}{\alpha_t} + \sqrt{2(\gamma+1)K+1}(u-l)\sqrt{\frac{\log t}{t}} + \frac{4\min(u-l, LK^{1/p}B)\alpha_t^K}{t^{(\gamma+1)K+1/2}}, \quad (10)$$

*and for $\alpha_t = \max(\lceil t^\gamma \rceil, 2)$ with $\gamma \ge 1/2$,*

$$\mathcal{R}_T^{DPDS}(f) \le 2(LK^{1/p}B + 4\min(u-l, LK^{1/p}B))\sqrt{T} + 2\sqrt{2(\gamma+1)K+1}(u-l)\sqrt{T\log T}. \quad (11)$$

Actually, we can relax the uniform Lipschitz continuity condition. Under the weaker condition of $|r(x^*) - r(x)| \le L\|x^* - x\|_p^q$ for all $x \in \mathcal{F}$ and for some constant $L > 0$, the incremental regret bound that is given in (10) becomes

$$\mathbb{E}(r(x^*) - r(x_{t+1}^{DPDS})) \le LK^{q/p}(B/\alpha_t)^q + (u-l)(\sqrt{2(\gamma+1)K+1}\sqrt{\log t / t} + 4\alpha_t^K t^{-(\gamma+1)K-1/2}).$$

The proof of Theorem 1 is derived by showing that the value of $x_{t+1}^* = \arg\max_{x \in \mathcal{F}_t} r(x)$ converges to the value of $x^*$ due to Lipschitz continuity, and the value of $x_{t+1}^{DPDS}$ converges to the value of $x_{t+1}^*$ via the use of concentration inequality inspired by [20, 17].

Even though the upper bound of regret in Theorem 1 depends on the budget $B$ linearly, this dependence can be avoided in the expense of increase in computational complexity. For example, in the literature, the reward is generally assumed to be in the unit interval, *i.e.*, $l = 0$ and $u = 1$, and the expected reward is assumed to be Lipschitz continuous with Euclidean norm and constant $L = 1$. In this case, by following the proof of Theorem 1, we observe that assigning $\gamma = 1/2$ and $\alpha_t = \max(\lceil \alpha t^\gamma \rceil, 2)$ for some $\alpha > 0$ gives a regret upper bound of $2B\sqrt{KT}/\alpha + 12\sqrt{KT\log T} + \alpha$ for $T > \alpha + 1$. Consequently, if $B = O(K)$, then $O(K^{3/4}\sqrt{T} + \sqrt{KT\log T})$ regret is achievable by setting $\alpha = K^{3/4}$.

### 3.4 Lower bound of regret for any bidding policy

We now show that DPDS in fact achieves the slowest possible regret growth. Specifically, Theorem 2 states that, for any bidding policy $\mu$ and horizon $T$, there exists a distribution $f$ for which the regret growth is slower than or equal to the square root of the horizon $T$.

**Theorem 2** *Consider the case where $K = 1$, $B = 1$, and $\lambda_t$ and $\pi_t$ are independent random variables with distributions*

$$f_\lambda(\lambda_t) = \epsilon^{-1} \mathbb{1}\{(1-\epsilon)/2 \leq \lambda_t \leq (1+\epsilon)/2\}$$

*and $f_\pi(\pi_t) = Bernoulli(\bar\pi)$, respectively. Let $f(\lambda_t, \pi_t) = f_\lambda(\lambda_t) f_\pi(\pi_t)$ and $\epsilon = T^{-1/2}/2\sqrt{5}$. Then, for any bidding policy $\mu$,*

$$R_T^\mu(f) \geq (1/16\sqrt{5})\sqrt{T},$$

*either for $\bar\pi = 1/2 + \epsilon$ or for $\bar\pi = 1/2 - \epsilon$.*

As seen in Theorem 2, we choose a specific distribution for the auction clearing and spot prices. Observe that, for this distribution, the payoff function is Lipschitz continuous with Lipschitz constant $L = 3/2$ because the magnitude of the derivative of the payoff function $|r'(x)| \leq |\bar\pi - x|/\epsilon \leq 3/2$ for $(1-\epsilon)/2 \leq x \leq (1+\epsilon)/2$ and $r'(x) = 0$ otherwise. So, it satisfies the condition given in Theorem 1.

The proof of Theorem 2 is obtained by showing that, every time the bid is cleared, an incremental regret greater than $\epsilon/2$ is incurred under the distribution with $\bar\pi = (1/2-\epsilon)$; otherwise, an incremental regret greater than $\epsilon/2$ is incurred under the distribution with $\bar\pi = (1/2+\epsilon)$. However, to distinguish between these two distributions, one needs $\Omega(T)$ samples, which results in a regret lower bound of $\Omega(\sqrt{T})$. The bound is obtained by adapting a similar argument used by [29] in the context of non-stochastic MAB problem.

## 4 Empirical study

New York ISO (NYISO), which consists of 11 zones, allows virtual transactions at zonal nodes only. So, we use historical DA and RT prices of these zones from 2011 to 2016 [30]. Since the price for each hour is different at each zone, there are $11 \times 24$ different locations, *i.e.*, zone-hour pairs, to bid on every day. The prices are per unit (MWh) prices. We also consider buy and sell bids simultaneously for all location. As explained in Sec. 1.1, a sell bid is a bid to sell in the DA market with an obligation to buy back in the RT market. Hence, the profit of a sell bid at period $t$ is $(\lambda_t - \pi_t)^\intercal \mathbb{1}\{x_t \leq \lambda_t\}$. Generally, an upper bound $\bar p$ for the DA prices is known, *e.g.* $\bar p = \$1000$ for NYISO. We convert a sell bid to a buy bid by using $x_t^{\text{sell}} = \bar p - x_t$, $\lambda_t^{\text{sell}} = \bar p - \lambda_t$, and $\pi_t^{\text{sell}} = \bar p - \pi_t$ instead of $x_t$, $\lambda_t$, and $\pi_t$. NYISO DA market for day $t$ closes at 5:00 am on day $t-1$. Hence, the RT prices of all hours of day $t-1$ cannot be observed before the bid submission for day $t$. Therefore, the most recent information used before the submission for day $t$ was the observations from day $t-2$.

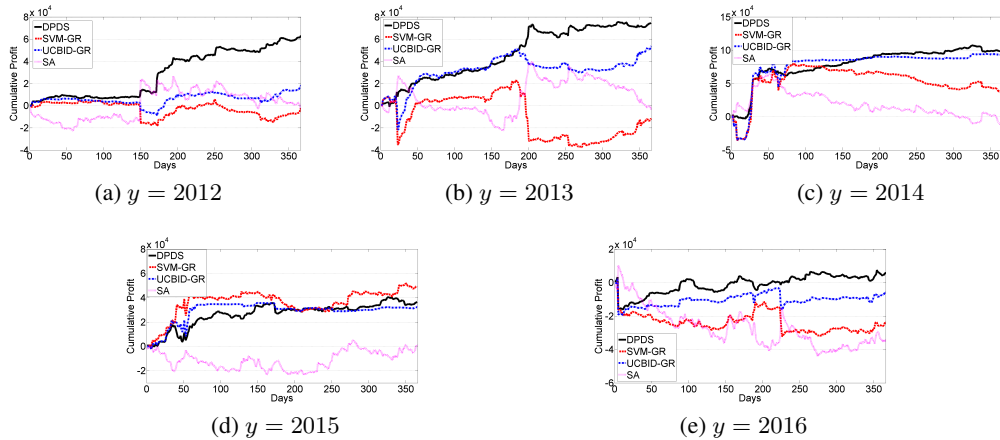

(a) $y = 2012$         (b) $y = 2013$         (c) $y = 2014$

(d) $y = 2015$         (e) $y = 2016$

Figure 3: Cumulative profit trajectory of year $y$ for $B = 100000$

We compare DPDS with three algorithms. One of them is UCBID-GR, inspired by UCBID [7]. At each day, UCBID-GR sorts all locations according to their profitabilities, *i.e.*, their price spread (the difference between DA and RT price) sample means. Then, starting from the most profitable location,

UCBID-GR sets the bid of a location equal to its RT price sample mean until there isn't any sufficient budget left.

The second algorithm, referred to as SA, is a variant of Kiefer-Wolfowitz stochastic approximation method. SA approximates the gradient of the payoff function by using the current observation and updates the bid of each $k$ as follows;

$$x_{t,k} = x_{t-1,k} + a_t \left( (\pi_{t-2,k} - \lambda_{t-2,k})(\mathbb{1}\{x_{t-1,k} + c_t \geq \lambda_{t-2,k}\} - \mathbb{1}\{x_{t-1,k} \geq \lambda_{t-2,k}\}) \right)/c_t.$$

Then, $x_t$ is projected to the feasible set $\mathcal{F}$.

The last algorithm is SVM-GR, which is inspired by the use of support vector machines (SVM) by Tang et al. [31] to determine if a buy or a sell bid is profitable at a location, *i.e.*, if the price spread is positive or negative. Due to possible correlation of the price spread at a location on day $t$ with the price spreads observed recently at that and also at other locations, the input of SVM for each location is set as the price spreads of all locations from day $t-7$ to day $t-2$. To test SVM-GR algorithm at a particular year, for each location, the data from the previous year is used to train SVM and to determine the average profit, *i.e.*, average price spread, and the bid level that will be accepted with 95% confidence in the event that a buy or a sell bid is profitable. For the test year, at each period, SVM-GR first determines if a buy or a sell bid is profitable for each location. Then, SVM-GR sorts all locations according to their average profits, and, starting from the most profitable location, it sets the bid of a location equal to the bid level with 95% confidence of acceptance until there isn't any sufficient budget left.

To evaluate the performance of a year, DPDS, UCBID-GR, and SA algorithms have also been trained starting from the beginning of the previous year. The algorithm parameter of DPDS was set as $\alpha_t = t$; and the step size $a_t$ and $c_t$ of SA were set as $20000/t$ and $2000/t^{1/4}$, respectively.

For B=\$100,000, the cumulative profit trajectory of five consecutive years are given in Fig. 3. We observe that DPDS obtains a significant profit in all cases, and it outperforms other algorithms consistently except 2015 where SVM-GR makes approximately 25% more profit. However, in three out of five years, SVM-GR suffers a considerable amount of loss. In general, UCBID-GR performs quite well except 2016, and SA algorithm incurs a loss almost every year.

## 5    Conclusion

By applying general techniques such as ERM, discretization approach, and dynamic programming, we derive a practical and efficient algorithm to the algorithmic bidding problem under budget constraint in repeated multi-commodity auctions. We show that the expected payoff of the proposed algorithm, DPDS, converges to that of the optimal strategy by a rate no slower than $\sqrt{\log t/t}$, which results in a $O(\sqrt{T \log T})$ regret. By showing that the regret is lower bounded by $\Omega(\sqrt{T})$ for any bidding strategy, we prove that DPDS is order optimal up to a $\sqrt{\log T}$ term.

For the motivating application of virtual bidding in electricity markets (see Sec. 1.1), the stochastic setting, studied in this paper, is natural due to the electricity markets being competitive, which implies that the existence of an adversary is very unlikely. However, it is also of interest to study the adversarial setting to extend the results to other applications. For example, the adversarial setting of our problem is a special case of no-regret learning problem of Simultaneous Second Price Auctions (SiSPA), studied by Daskalakis and Syrgkanis [32] and Dudik et al. [33].

In particular, to deal with the adversarial setting, it is possible to use our dynamic programming approach as the offline oracle for the Oracle-Based Generalized FTPL algorithm proposed by Dudik et al. [33] if we fix the discretized action set over the whole time horizon. More specifically, let the interval length of discretization be $B/m$, *i.e.*, $\alpha_t = m$. Then, it is possible to show that a 1-admissible translation matrix with $K\lceil \log m \rceil$ columns is implementable with complexity $m$. Consequently, no-regret result of Dudik et al. [33] holds with a regret bound of $O(K\sqrt{T} \log m)$ if we measure the performance of the algorithm against the best action in hindsight in the discretized finite action set rather than in the original continuous action set considered here. Unfortunately, as shown by Weed et al. [7], it is not possible to achieve sublinear regret with a fixed discretization for the specific problem considered in this paper. Hence, it requires further work to see if this method can be extended to obtain no-regret learning for the adversarial setting under the original continuous action set.

**Acknowledgments**

We would like to thank Professor Robert Kleinberg for the insightful discussion.

This work was supported in part by the National Science Foundation under Award 1549989 and by the Army Research Laboratory Network Science CTA under Cooperative Agreement W911NF-09-2-0053.

## Footnotes

[1]This implies that the auction clearing price is independent of bid $x_t$, which is a reasonable assumption for any market where an individual's bid has negligible impact on the market price.

[2]For example, transmission congestion may prevent scheduling the least expensive resources at some locations.

[3]This is reasonable in the case of virtual trading because DA and RT prices are bounded due to offer/bid caps.

[4]The regret definition used here is the same as in [14]. This definition is also known as pseudo-regret in the literature [25].

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
