[Supplementary Material]

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

# A  Simulation study

Here, we present a simulation example to illustrate the regret growth rate of DPDS. We consider an example with $K = 5$. In this example, $\pi_t$ and $\lambda_t$ are independent, $\lambda_t$ is exponentially distributed with mean $\bar{\lambda} = [4, 6, 8, 8, 4]^{\mathsf{T}}$, and $\pi_t$ is uniformly distributed with mean $\bar{\pi} = [5, 8, 8, 9, 3]^{\mathsf{T}}$ and support in $[\bar{\pi} - 1, \bar{\pi} + 1]$. Previously, in Sec. 2.1, we stated the characterization of the optimal solution for this example. By using this characterization, we determined the optimal solution and the associated budget $B$ for a range of values of the Lagrange multiplier $\gamma^*$ of the budget constraint. More specifically, for the values $0.1, 0.2, 0.3$, and $0.4$ of $\gamma^*$, the corresponding values of $B$ are $25.828, 20.870, 17.018, 13.845$, respectively. We evaluate the performance of algorithms for these four different values of $B$.

(a) Regret when $B = 13.845$      (b) Regret when $B = 17.018$

(c) Regret when $B = 20.870$      (d) Regret when $B = 25.828$

Figure 4: Regret with respect to $\sqrt{t}$

As a benchmark comparison we consider two different approaches. The first one is based on a sliding window (SW) forecasting approach that calculates the average payoff function of each good every day from the prices of last ten days only. Then, it determines the optimal solution maximizing the total average payoff by solving the integer linear program given in (7). The second one, referred to as SA, is a variant of Kiefer-Wolfowitz stochastic approximation method as explained in Sec. 4. Recall that SA approximates the gradient of the payoff function using the current observation and updates the bid of each $k$ as follows;

$$x_{t+1,k} = x_{t,k} + a_t \left( (\pi_{t,k} - \lambda_{t,k})(\mathbb{1}\{x_{t,k} + c_t \geq \lambda_{t,k}\} - \mathbb{1}\{x_{t,k} \geq \lambda_{t,k}\}) \right) / c_t.$$

Then, SA projects $x_{t+1}$ to the feasible set $\mathcal{F}$. To give a good result for $B = 13.845$, step size $a_t$ and $c_t$ were carefully chosen to be $5.5/t$ and $2.5/t^{1/4}$, respectively. We set the DPDS algorithm parameter $\alpha_t = t$.

To calculate the average performance, 1000 Monte Carlo runs were used. The regret performances for budgets $13.845, 17.018, 20.870$ and $25.828$ are given in Fig. 4. In all cases, DPDS outperforms, and its order of regret growth is actually better than $\sqrt{T}$. When the SA algorithm parameters are tuned well, we observe that its performance may get close to DPDS as in Fig. 4a. However, when we increase the budget to $25.828$ gradually, the performance of SA deteriorates significantly. Also, as seen in Fig. 4, the regret of SW is much higher than DPDS and SW because SW does not converge to the optimal solution due to fixed number of samples used in prediction.

# B DPDS algorithm pseudo-code

---

**Algorithm 1** DPDS policy

---

1: **Input:** $x_1 = 0$; Initialize $\left(\lambda^{(k)}, r^{(k)}\right) = (0,0) \; \forall \; k \in \{1,...,K\}$;
2: **for** $t = 1$ to $T$ **do**
3:     Bid $x_t$;
4:     At the end of period $t$, observe $(\lambda_t, \pi_t)$ and update $\left(\lambda^{(k)}, r^{(k)}\right) \; \forall \; k \in \{1,...,K\}$ using (5) given in Sec. 3.1;
5:     Set $V_0(jB/\alpha_t) = 0 \; \forall \; j \in \{0, 1, ..., \alpha_t\}$ and $V_n(0) = 0 \; \forall \; n \in \{1, ..., K\}$;
6:     Set $w_n(0) = 0 \; \forall \; n \in \{1, ..., K\}$;
7:     **for** $n = 1$ to $K$ **do**
8:       $l = 1$, $d = 0$, and $j' = \alpha_t$;
9:       **for** $j = 1$ to $\alpha_t$ **do**
10:         **while** $d = 0$ **do**
11:           **if** $\lambda_l^{(n)} > jB/\alpha_t$ **then**
12:             $\hat{r}_{t,n}(jB/\alpha_t) = r_{l-1}^{(n)}$
13:             break;
14:           **else**
15:             **if** $l = t + 1$ **then**
16:                $\hat{r}_{t,n}(jB/\alpha_t) = r_{t+1}^{(n)}$;
17:                $d = 1$ and $j' = j$;
18:                break;
19:             **else**
20:                $l = l + 1$;
21:             **end if**
22:           **end if**
23:         **end while**
24:         $V_n(jB/\alpha_t) = V_{n-1}(jB/\alpha_t)$ and $w_n(jB/\alpha_t) = 0$;
25:         **for** $i = 1$ to $\min\{j, j'\}$ **do**
26:           **if** $V_n(jB/\alpha_t) < V_{n-1}((j-i)B/\alpha_t) + \hat{r}_{t,n}(iB/\alpha_t)$ **then**
27:             $V_n(jB/\alpha_t) = V_{n-1}((j-i)B/\alpha_t) + \hat{r}_{t,n}(iB/\alpha_t)$;
28:             $w_n(jB/\alpha_t) = iB/\alpha_t$;
29:           **end if**
30:         **end for**
31:       **end for**
32:     **end for**
33:     $B_r = B$;
34:     **for** $k = K$ to $1$ **do**
35:       $x_{t+1,k} = w_k(B_r)$;
36:       $B_r = B_r - x_{t+1,k}$;
37:     **end for**
38: **end for**

---

# C Proof of Theorem 1

Recall that $x^* = \arg\max_{x \in \mathcal{F}} r(x)$ and let $x_{t+1}^* = \arg\max_{x \in \mathcal{F}_t} r(x)$. Hence, for any $x' \in \mathcal{F}_t$,

$$r(x^*) - r(x_{t+1}^*) \leq r(x^*) - r(x').$$

We take $x_k' = \lfloor x_k^*/(B/\alpha_t) \rfloor (B/\alpha_t)$ for all $k \in \{1, ..., K\}$, where $\lfloor x_k^*/(B/\alpha_t) \rfloor$ denotes the largest integer smaller or equal to $x_k^*/(B/\alpha_t)$, so that $x' \in \mathcal{F}_t$ and $|x_k' - x_k^*| \leq B/\alpha_t$ for all $k \in \{1, ..., K\}$. Then, due to Lipschitz continuity of the expected payoff function $r(.)$ on $\mathcal{F}$ with p-norm and constant $L$,

$$r(x^*) - r(x_{t+1}^*) \leq LK^{1/p}B/\alpha_t. \tag{12}$$

Since the payoff obtained at each period $t$ is in $[l, u]$ and $r(.)$ is Lipschitz, $r(x_{t+1}^*) - r(x) \leq C$ for any $x \in \mathcal{F}_t$ where $C = \min(u - l, LK^{1/p}B)$. Then, for any $\delta_t > 0$,

$$r(x_{t+1}^*) - r(x_{t+1}^{\text{DPDS}}) = \sum_{x \in \mathcal{F}_t} (r(x_{t+1}^*) - r(x)) \mathbb{1}\{x_{t+1}^{\text{DPDS}} = x\}$$

$$\leq \delta_t \sum_{x \in \mathcal{F}_t : r(x_{t+1}^*) - r(x) \leq \delta_t} \mathbb{1}\{x_{t+1}^{\text{DPDS}} = x\} + C \sum_{x \in \mathcal{F}_t : r(x_{t+1}^*) - r(x) > \delta_t} \mathbb{1}\{x_{t+1}^{\text{DPDS}} = x\}$$

$$\leq \delta_t + C \sum_{x \in \mathcal{F}_t : r(x_{t+1}^*) - r(x) > \delta_t} \mathbb{1}\{x_{t+1}^{\text{DPDS}} = x\},$$

where the last inequality is obtained by the fact that at most one of the indicator functions can be equal to 1 due to the events being disjoint. For any $x \in \mathcal{F}_t$, $\hat{r}_t(x) \geq \hat{r}_t(x_{t+1}^*)$ has to hold if $x_{t+1}^{\text{DPDS}} = x$ holds. Hence, we can upper bound the last inequality obtained to get

$$r(x_{t+1}^*) - r(x_{t+1}^{\text{DPDS}}) \leq \delta_t + C \sum_{x \in \mathcal{F}_t : r(x_{t+1}^*) - r(x) > \delta_t} \mathbb{1}\{\hat{r}_t(x) \geq \hat{r}_t(x_{t+1}^*)\}.$$

Now, observe that for $\hat{r}_t(x) \geq \hat{r}_t(x_{t+1}^*)$ to hold for any $x \in F_t$ satisfying $r(x_{t+1}^*) - r(x) > \delta_t$, the event

$$\mathcal{E}_1 = \{\hat{r}_t(x_{t+1}^*) + \delta_t/2 \leq r(x_{t+1}^*)\}$$

holds and/or the event

$$\mathcal{E}_2 = \{r(x) + \delta_t/2 \leq \hat{r}_t(x)\}$$

holds. Consequently,

$$\mathbb{E}(r(x_{t+1}^*) - r(x_{t+1}^{\text{DPDS}})) \leq \delta_t + C \sum_{x \in \mathcal{F}_t : r(x_{t+1}^*) - r(x) > \delta_t} \Pr(\mathcal{E}_1 \cup \mathcal{E}_2).$$

For any fixed value of $x \in \mathcal{F}$, $\{(\pi_i - \lambda_i)^\intercal \mathbb{1}\{x \geq \lambda_i\}\}_{i=1}^t$ are i.i.d. random variables taking values in $[l, u]$ with mean $r(x)$. Therefore, by Hoeffding's inequality, both $\Pr(\mathcal{E}_1)$ and $\Pr(\mathcal{E}_2)$ are upper bounded by $\exp\{-t\delta_t^2/(2(u-l)^2)\}$. Using the fact that the cardinality of the set $\{x \in \mathcal{F}_t : r(x_{t+1}^*) - r(x) > \delta_t\}$ is upper bounded by $\alpha_t^K + K \leq 2\alpha_t^K$ for $\alpha_t \geq 2$ and $\Pr(\mathcal{E}_1 \cup \mathcal{E}_2) \leq \Pr(\mathcal{E}_1) + \Pr(\mathcal{E}_2)$, we get

$$\mathbb{E}(r(x_{t+1}^*) - r(x_{t+1}^{\text{DPDS}})) \leq \delta_t + 4C\alpha_t^K \exp\{-t\delta_t^2/(2(u-l)^2)\}. \tag{13}$$

By using (12) and (13) and setting $\delta_t = \sqrt{2(\gamma+1)K + 1}(u-l)\sqrt{\log t/t}$, we obtain

$$\mathbb{E}(r(x^*) - r(x_{t+1}^{\text{DPDS}})) = \mathbb{E}(r(x^*) - r(x_{t+1}^*)) + \mathbb{E}(r(x_{t+1}^*) - r(x_{t+1}^{\text{DPDS}}))$$
$$\leq LK^{1/p}B/\alpha_t + \sqrt{2(\gamma+1)K + 1}(u-l)\sqrt{\log t/t} + 4C\alpha_t^K t^{-(\gamma+1)K-1/2}.$$

For any $T \geq 2$, $\sum_{t=1}^{T-1} 1/\sqrt{t} \leq 2\sqrt{T-1} - 1$ and $\sum_{t=1}^{T-1} \sqrt{\log t/t} \leq 2\sqrt{(T-1)\log(T-1)}$. Hence, for any $\alpha_t = \max(\lceil t^\gamma \rceil, 2)$ with $\gamma \geq 1/2$ and $T > 2$,

$$\sum_{t=2}^{T-1} \mathbb{E}(r(x^*) - r(x_{t+1}^{\text{DPDS}})) \leq \left(LK^{1/p}B + 4C\right) \sum_{t=1}^{T-1} \frac{1}{\sqrt{t}} + \sqrt{2(\gamma+1)K + 1}(u-l) \sum_{t=1}^{T-1} \sqrt{\frac{\log t}{t}}$$

$$\leq \left(LK^{1/p}B + 4C\right)\left(2\sqrt{T-1} - 1\right)$$
$$+ 2\sqrt{2(\gamma+1)K + 1}(u-l)\sqrt{(T-1)\log(T-1)}.$$

Since $\mathbb{E}(r(x^*) - r(x_t^{\text{DPDS}})) \leq C$, for any $T \geq 1$,

$$\mathcal{R}_T^{\text{DPDS}}(f) \leq 2(LK^{1/p}B + 4C)\sqrt{T} + 2\sqrt{2(\gamma+1)K + 1}(u-l)\sqrt{T\log T}.$$

∎

## D Proof of Theorem 2

Fix any policy $\mu$. Since $\lambda_t$ and $\pi_t$ are independent,

$$r(x) = \mathbb{E}((\bar{\pi} - \lambda_t)\mathbb{1}\{x \geq \lambda_t\}|x)$$

and

$$r(x^*) - r(x_t^\mu) = \mathbb{E}((\bar{\pi} - \lambda_t)(\mathbb{1}\{x^* \geq \lambda_t\} - \mathbb{1}\{x_t^\mu \geq \lambda_t\})|x_t^\mu, x^*) \tag{14}$$

Let $f_0$, $f_1$, $f_2$ denote the distribution of $\{\lambda_t, \pi_t\}_{t=1}^T$ and policy $\mu$ under the choice of $\bar{\pi} = 1/2$, $\bar{\pi} = 1/2 - \epsilon$, and $\bar{\pi} = 1/2 + \epsilon$, respectively. Also, let $\mathbb{E}_i(.)$ and $\mathcal{R}_T^\mu(f_i)$ denote the expectation with respect to the distribution $f_i$ and the regret of policy $\mu$ under distribution $f_i$, respectively.

Under distribution $f_1$, observe that $\bar{\pi} - \lambda_t \leq -\epsilon/2$ for any value of $\lambda_t$. Therefore, optimal solution under known distribution $x^* \in [0, (1-\epsilon)/2]$ so that $\mathbb{1}\{x^* \geq \lambda_t\} = 0$. Then, by using (14), the regret given in (3) in Sec. 2.1 can be expressed as

$$\mathcal{R}_T^\mu(f_1) = \mathbb{E}_1\left(\sum_{t=1}^T -(\bar{\pi} - \lambda_t)\mathbb{1}\{x_t^\mu \geq \lambda_t\}\right) \geq \frac{\epsilon}{2}\mathbb{E}_1\left(\sum_{t=1}^T \mathbb{1}\{x_t^\mu \geq \lambda_t\}\right).$$

Similarly, under distribution $f_2$, observe that $\bar{\pi} - \lambda_t \geq \epsilon/2$ for any value of $\lambda_t$. Therefore, optimal solution under known distribution $x^* \in [(1+\epsilon)/2, 1]$ so that $\mathbb{1}\{x^* \geq \lambda_t\} = 1$. Then, by using (14), the regret can be expressed as

$$\mathcal{R}_T^\mu(f_2) = \mathbb{E}_2\left(\sum_{t=1}^T (\bar{\pi} - \lambda_t)\mathbb{1}\{x_t^\mu < \lambda_t\}\right) \geq \frac{\epsilon}{2}\mathbb{E}_2\left(\sum_{t=1}^T \mathbb{1}\{x_t^\mu < \lambda_t\}\right).$$

For any non-negative bounded function $h$ defined on information history $I_T = \{x_t, \lambda_t, \pi_t\}_{t=1}^T$ such that $0 \leq h(I_T) \leq M$ for some $M \geq 0$ and for any distributions $p$ and $q$, the difference between the expected value of $h$ under the distributions $p$ and $q$ is bounded by a function of the KL-divergence between these distributions as follows:

$$\mathbb{E}_q(h(I_T)) - \mathbb{E}_p(h(I_T)) \leq \int_{q(I_T)>p(I_T)} h(I_T)(q(I_T) - p(I_T))dI_T$$

$$\leq M \int_{q(I_T)>p(I_T)} q(I_T) - p(I_T)dI_T$$

$$= M\frac{1}{2}\int |q(I_T) - p(I_T)|dI_T$$

$$\leq M\sqrt{\mathrm{KL}(q||p)/2}. \tag{15}$$

where $\mathrm{KL}(q||p) = \int q(I_T)\log(q(I_T)/p(I_T))dI_T$ is the KL-divergence between $q$ and $p$ and the last inequality is due to Pinsker's inequality [34], *i.e.*, $V(q,p) \leq \sqrt{\mathrm{KL}(q||p)/2}$ where $V(q,p) = \int |q(I_T) - p(I_T)|dI_T/2$ is the variational distance between $q$ and $p$. The bound given in (15) is inspired by a similar bound obtained by Auer et al. [29] in the proof of Lemma A.1 for the case of discrete distribution in the context of non-stochastic multi-armed bandit problem.

Now, since $\sum_{t=1}^T \mathbb{1}\{x_t^\mu \geq \lambda_t\} \leq T$ and $\sum_{t=1}^T \mathbb{1}\{x_t^\mu < \lambda_t\} \leq T$, we use (15) to obtain

$$\mathcal{R}_T^\mu(f_1) \geq \frac{\epsilon}{2}\left(\mathbb{E}_0\left(\sum_{t=1}^T \mathbb{1}\{x_t^\mu \geq \lambda_t\}\right) - T\sqrt{KL(f_0||f_1)/2}\right),$$

and

$$\mathcal{R}_T^\mu(f_2) \geq \frac{\epsilon}{2}\left(\mathbb{E}_0\left(\sum_{t=1}^T \mathbb{1}\{x_t^\mu < \lambda_t\}\right) - T\sqrt{KL(f_0||f_2)/2}\right).$$

Consequently,

$$\max_{i\in\{1,2\}} \mathcal{R}_T^\mu(f_i) \geq \frac{1}{2}\left(\mathcal{R}_T^\mu(f_1) + \mathcal{R}_T^\mu(f_2)\right)$$

$$\geq \frac{\epsilon}{4}\left(T - T\sqrt{\mathrm{KL}(f_0||f_1)/2} - T\sqrt{\mathrm{KL}(f_0||f_2)/2}\right). \tag{16}$$

For any $i \in \{0, 1, 2\}$, we can express the distribution of observations in terms of conditional distributions as follows;

$$f_i(I_T) = \prod_{t=1}^{T} f_i(\pi_t, \lambda_t | x_t^\mu, I_{t-1}) f_i(x_t^\mu | I_{t-1})$$

$$= \prod_{t=1}^{T} f_i(\pi_t) f_\lambda(\lambda_t) f(x_t^\mu | I_{t-1}),$$

where the second equality is due to the independence of $\lambda_t$ and $\pi_t$ from the past observations $I_{t-1}$, the bid $x_t^\mu$, and from each other. Also, the distribution of $x_t^\mu$ given $I_{t-1}$ does not depend on $i$. Consequently, for $i \in \{1, 2\}$,

$$\mathrm{KL}(f_0 \| f_i) = \int f_0(I_T) \log \left( \prod_{t=1}^{T} \frac{f_0(\pi_t)}{f_i(\pi_t)} \right) dI_T$$

$$= \sum_{t=1}^{T} \int f_0(I_T) \log \left( \frac{f_0(\pi_t)}{f_i(\pi_t)} \right) dI_T$$

$$= \sum_{t=1}^{T} \left( \frac{1}{2} \log \left( \frac{1/2}{1/2 + \epsilon} \right) + \frac{1}{2} \log \left( \frac{1/2}{1/2 - \epsilon} \right) \right)$$

$$= -(T/2) \log \left( 1 - 4\epsilon^2 \right).$$

Then, by (16) and by setting $\epsilon = T^{-1/2}/2\sqrt{5}$, we get

$$\max_{i \in \{1,2\}} \mathcal{R}_T^\mu(f_i) \geq \frac{\epsilon T}{4} \left( 1 - \sqrt{-T \log \left( 1 - 4\epsilon^2 \right)} \right)$$

$$= \frac{\sqrt{T}}{8\sqrt{5}} \left( 1 - \sqrt{-T \log \left( 1 - 1/(5T) \right)} \right)$$

$$\geq \frac{\sqrt{T}}{16\sqrt{5}}$$

where the last inequality follows from the fact that $-\log(1 - x) \leq (5/4)x$ for $0 \leq x \leq 1/5$. $\blacksquare$