[Reviews · NeurIPS 2017]

Reviewer 1



The paper presents a simple yet interesting problem. That of optimizing a bidder payoff in repeated auctions. The setup considered by the paper is one where payouts and spot prices are random. The learner must choose his bidding strategy under a budget constrained. While in principle this problem can be solved doing empirical payoff maximization, as the authors show, this is in fact NP hard. The contribution of the paper is to provide a dynamic program solution to the empirical optimization problem under the reasonable assumption of payoffs being Lipchitz in expectation as a function of the bids. The algorithm is simple and straightforward, while the proofs are fairly standard. Nevertheless the authors are still the first to propose this solution to the problem. The empirical results are also compelling. On the down side, since the authors are considering a stochastic setup it would be interesting to measure the performance of this algorithm in terms of a notion of pseudo-regret in which case one would expect it to be constant. Finally, the main issue with this paper is its lower bound. Indeed, the lower bound provided here corresponds to an adversarial setup. It is the usual random distribution that depends on the number of rounds chosen to prove lower bounds in bandits and expert learning algorithms in an adversarial scenario.

Reviewer 2



*I have read the rebuttal.* Summary: This paper studies online repeated bidding strategy for a specific multi-commodity auction, where at each time the bidder needs to decide how to allocate a fixed budget among K goods with individual clearing price and utility. The environment is assumed to be iid so that a simple follow the leader approach will admit sublinear regret. The difficulty is that computing ERM is NP-hard for this problem and the paper thus proposes to discretize the possible bids so that a simple dynamic programming approach can efficiently compute the ERM. A regret bound of order \sqrt{T} is proven under some Lipschitz continuity assumption, and a matching lower bound is also provided. The paper is concluded with an experiment on real data comparing several algorithms. Major Comments: Overall the problem is quite interesting and the experimental results are promising. The technical contributions of the paper is not very strong though. Everything appears to be straightforward: the environment is stochastic so follow the leader works already; computing exact ERM is NP-hard, so discretization is a natural idea; the dynamic programming formula is also not difficult to come up with. One suggestion I have is to see whether techniques from a recent work of Dudik et al. "Oracle-Efficient Online Learning and Auction Design", FOCS 2017 can be used here to deal with adversarial setting. Their approach assumes an ERM oracle, which is exactly provided by this work. Minor Comments: 1. The paper keeps mentioning related works on MAB, but the setting here is full information instead of bandit. I think the related works discussion could be improved in this sense. 2. Eq.(1), maybe define what \geq mean for vectors. 3. L146, the solution of the example needs more explanation. 4. L172, r'_{k,i}, maybe swap the two subscripts to make notation more consistent. 5. In the definition of V_n(b), I think \hat{r}_t should be \hat{r}_{t,i}.

Reviewer 3



This paper studies the online learning (stochastic and full-information) problem of bidding in multi commodity first price auctions. The paper introduces a polynomial time algorithm that achieves a regret of \sqrt{T log(T)} that has a near optimal dependence on T. The main challenge that the paper has to deal with is to find a computationally efficient algorithm for computing the best biding strategy given a known distribution.The authors first demonstrate that natural approaches for solving this problem exactly are not computationally efficient (this is not a formal np-hardness proof). Then, they provide a FPTAS for solving the problem using dynamic programming. Once they have a FPTAS for the offline problem, their results hold for the stochastic online setting using existing reductions. I haven’t carefully looked in to the details of their analysis of the dynamic programming, but I think the effectiveness of it here is interesting and surprising — specially given that the variation of this problem for the second price auctions is hard to approximate. As for the weaknesses, I find the theorems to be hard to interpret. Particularly because there are a lot of variables in them. The authors should provide one simple variation of theorem 1 with fewer parameters. For example, when l = 0, and u = 1, the lipschitz conditions holds with L = 1 and p = 1, and B = K. Assigning \alpha = \sqrt{t} gives a regret of \tilde O(\sqrt{t \log(T)}). As mentioned above the results mentioned here hold for the full-information online stochastic setting, that is, the valuation and the clearing price are drawn from a the same distribution at every step. I find this to be a weird choice of a setting. In most cases, full information settings are accompanied by adversarial arrival of the actions, rather than stochastic. This is indeed due to the fact that the statistical and computational aspects of online stochastic full-information setting is very similar to the offline stochastic full-information setting. So, I think a much more natural setting would have been to consider the adversarial online setting. In this regard, there are two very relevant works on no-regret learning in auctions that should be cited here. Daskalakis and Syrgkanis FOCS’16 and later on Dudik et al. FOCS’17 study the problem of online learning in Simultaneous Second Price auctions (SiSPA). This is also the problem of optimal bidding in a repeated multi-commodity auction where each item is assigned based on a second price auctions, rather than a first price auction that is considered in this paper. In particular, the latter paper discusses computationally efficient mechanisms for obtaining a no-regret algorithm in the adversarial setting when one can solve the offline problem efficiently. Their methods seem to apply to the problem discussed in this paper and might be specially useful since this paper provides an approximation method for solving the offline problem. It is worth seeing whether the machinery used in their paper can directly strengthen your result so that it holds for the adversarial online setting. After rebuttal: I suggest that the authors take a closer look at the works of Dudik et al. and Daskalakis and Syrgkanis. In their response, the authors mention that those works are inherently for discrete spaces. From a closer look at the work of Dudik et al. it seems that they also work with some auctions specifically in the continuous space by first showing that they can discretize the auction space. This discretization step is quite common in no-regret learning in auctions. Given that the current submission also shows that one can discretize the auctions space first, I think the results of Dudik et al. can be readily applied here. This would strengthen the results of this paper and make the setting more robust. I suggest that the authors take a closer look at these works when preparing their discussion.